# Molecular Changes in Dengue Envelope Protein Domain III upon Interaction with Glycosaminoglycans

**DOI:** 10.3390/pathogens9110935

**Published:** 2020-11-11

**Authors:** James G. Hyatt, Sylvain Prévost, Juliette M. Devos, Courtney J. Mycroft-West, Mark A. Skidmore, Anja Winter

**Affiliations:** 1School of Life Sciences, Keele University, Huxley Building, Keele, Staffordshire ST5 5BG, UK; j.g.hyatt@keele.ac.uk (J.G.H.); c.j.mycroft-west@keele.ac.uk (C.J.M.-W.); m.a.skidmore@keele.ac.uk (M.A.S.); 2Large Scale Structures Group, Institut Laue-Langevin, 71 avenue des Martyrs, CS 20156, 38042 Grenoble CEDEX 9, France; prevost@ill.fr; 3Life Sciences Group, Institut Laue-Langevin, 71 avenue des Martyrs, CS 20156, 38042 Grenoble CEDEX 9, France; devosj@ill.fr

**Keywords:** dengue virus, envelope protein, glycosaminoglycans, protein-glycosaminoglycan interactions, small-angle scattering

## Abstract

Dengue fever is a rapidly emerging vector-borne viral disease with a growing global burden of approximately 390 million new infections per annum. The Dengue virus (DENV) is a flavivirus spread by female mosquitos of the *aedes* genus, but the mechanism of viral endocytosis is poorly understood at a molecular level, preventing the development of effective transmission blocking vaccines (TBVs). Recently, glycosaminoglycans (GAGs) have been identified as playing a role during initial viral attachment through interaction with the third domain of the viral envelope protein (EDIII). Here, we report a systematic study investigating the effect of a range of biologically relevant GAGs on the structure and oligomeric state of recombinantly generated EDIII. We provide novel in situ biophysical evidence that heparin and chondroitin sulphate C induce conformational changes in EDIII at the secondary structure level. Furthermore, we report the ability of chondroitin sulphate C to bind EDIII and induce higher-order dynamic molecular changes at the tertiary and quaternary structure levels which are dependent on pH, GAG species, and the GAG sulphation state. Lastly, we conducted ab initio modelling of Small Angle Neutron Scattering (SANS) data to visualise the induced oligomeric state of EDIII caused by interaction with chondroitin sulphate C, which may aid in TBV development.

## 1. Introduction

Dengue virus (DENV) causes the world’s most prevalent mosquito-borne flavivirus (FLV) disease, placing 3.9 billion people at risk of infection and resulting in approximately 20 million cases each year in 129 countries around the tropics and subtropics [1,2]. FLVs are enveloped, positive single-stranded RNA viruses causing varying symptoms from haemorrhagic fever and fatal neurological diseases to foetal defects. To date, there is no completely effective vaccine available for treatment against many FLVs [3], and in light of recent outbreaks of FLV around the world—most notably, the Zika outbreak in South America in 2015—much effort has therefore been made to understand the molecular mechanism by which FLVs such as Dengue infect host cells in a bid to identify an effective treatment or transmission blocking vaccine (TBV).

The mechanism by which DENV infects its target host cell is the major determinant of the virus cellular tropism and is critical for viral pathogenesis. Glycosaminoglycans (GAGs) are thought to be the initial co-receptors that all pathogenic FLVs utilise for the infection of host cells [4,5,6], which is followed by interaction with protein-based receptors [7]. Finally, FLVs infiltrate the host cell through clathrin-mediated or independent endocytosis, accompanied by a pH-dependent conformational change in the envelope protein, and followed by membrane fusion and the release of the viral genome into the cell [8].

GAGs are anionic, unbranched polysaccharides composed of repeating disaccharide units located on the surface of eukaryotic cells and in their extracellular matrix (ECM). They differ according to the type of hexosamine, hexose, or hexuronic acid unit that they contain; their sulphation state; and the chirality and position of the glycosidic linkage between their saccharide units [9,10]. Additionally, GAGs may harbour various sulphation patterns, allowing for an increased variability of interaction partners, with certain GAG–protein interactions requiring sulphation at specific points within both the monomeric carbon ring as well as the polysaccharide sequence. For instance, heparan sulphate (HS) is less sulphated than heparin [11] (for a structural overview of the different GAGs, see Appendix A). As such, HS displays more structural heterogeneity with regards to the sulphation state and patterns amongst dimeric subunits. This allows for increased variability in its spatial conformation, resulting in a more varied range of potential interaction partners, including growth factors, morphogens, ECM proteins, and pathogen surface proteins [12,13]. For instance, HS has been shown to serve as a receptor or co-receptor for the DENV infection of host cells [12,13], and GAG knockout studies have demonstrated a direct correlation between the presence of GAGs and the infectivity of different Dengue subtypes [4,14]. The type of GAG and its degree of sulphation influences the binding activity and infection rate of different FLVs [15], with many lesser sulphated variants of HS being unable to bind viral envelope proteins entirely [13,16].

The flaviviral envelope protein (E protein) forms the first contact with a host cell and potential receptors and thus initiates infection [8]. The Dengue E protein consists of three domains (I–III) which undertake different tasks in the infection process [8,17,18]. pH-dependent conformational changes are important in the fusion event, as they alter the spatial arrangement of DENV E protein domains and open important ligand binding pockets [17]. Particularly, the hairpin between β-strands “k” and “l” at the interface between domains I and II has been identified as a key structural element for initiating the low-pH conformational change that leads to the formation of fusion-competent DENV envelope protein trimmers [8,17]. During membrane fusion, the third domain (EDIII) shifts and rotates, then folds back onto domain I and thus facilitates final trimer interaction [8].

It is generally thought that FLV envelope proteins engage host receptor proteins and contain GAG binding sites comprised of many basic, acidic, and polar residues that contribute to binding [19,20,21,22,23]. DENV EDIII in particular contains four residues (382–385) between ß-strands F and G which form a compact solvent-exposed extended loop motif that has been implicated in receptor binding [24]. This FG loop was shown to be critical for the infection of *Aedes aegypti* mosquito midguts and mammalian cells by mutational studies [25]. However, no study at the molecular level has been performed to identify the specific residues involved in GAG binding.

To date, there has been no systematic study of the interaction between EDIII and different biologically relevant GAGs investigating the dynamic changes that occur upon binding. The role of GAGs in initial host receptor–viral envelope protein binding is a poorly understood part of viral endocytosis. Therefore, we set out to investigate these interactions using biophysical techniques such as circular dichroism (CD) and small angle scattering (SAXS and SANS) in order to better understand the importance of GAGs in the infection process.

We report here distinct dynamic changes at a molecular level that occur when different GAGs bind EDIII at different pHs. We show that heparin induces subtle changes in EDIII at the secondary structure level, whereas chondroitin sulphate C (CSC) induces changes at both the molecular as well as the super-molecular level. We observe a pH- and sulphation state-dependent formation of higher-order oligomers of EDIII in the presence of CSCi and modelled these as three to five EDIII molecules interacting with a linear CSCi molecule.

Our findings provide evidence for binding of the Dengue envelope protein to GAGs and will be invaluable in understanding molecular interactions occurring in the initial phase of host recognition by the virus, which is important for developing effective TBVs that target this initial interaction event.

## 2. Materials and Methods

### 2.1. Protein Expression from E. coli

A synthetic clone was generated encompassing the nucleotide sequences encoding for EDIII of DENV2 (strain Jamaica/1409/1983, amino acids 296–394, Appendix A) with an N-terminal 6x-histidine tag and tobacco etch virus (TEV) cleavage sequence into pRSET_A (Genewiz, South Plainfield, NJ, USA), and transformed into *Escherichia coli* BL21(DE3) pLysS cells (Promega, Dane County, WI, USA). Cells were grown in LB media supplemented with 100 µg/mL of ampicillin and 25 µg/mL of chloramphenicol, and expression was induced at an OD_600_ of 0.6 using 0.5 mM of IPTG (isopropyl β-D-1-thiogalactopyranoside, final concentration) for 24 h at 37 °C. Cells were harvested at 3500× *g* for 30 min, and the pellet was resuspended in PBS (phosphate buffered saline) and frozen at −20 °C.

### 2.2. Protein Expression in E. coli Using Matchout Deuterated Minimal Medium

Small angle neutron scattering is a structural technique sensitive to isotopes, and specifically to a contrast between 1H (Hydrogen) and 2H or D (Deuterium). At their natural isotopic composition (hydrogenous material), proteins and GAGs have very similar contrast for neutrons. To selectively see the structural modifications of EDIII and render the GAGs “transparent”, we used isotopically labelled EDIII protein (85% D) produced in the Deuteration Laboratory of the Institut Laue-Langevin (ILL, Grenoble, France). A transposition reaction was carried out on the original pRSET_A vector to modify the resistance selection marker from ampicillin to kanamycin using the EZ-Tn5™ <KAN-2> Insertion Kit transposition kit (EZ1982K, Lucigen^®^, Middleton, WI, USA). New kanamycin-resistant pRSET_A EDIII vector was transformed into One Shot™BL21(DE3) *E. coli* (Invitrogen, Carlsbad, CA, USA). These cells were adapted to deuterated Enfors minimal medium [26] at 37 °C in the presence of kanamycin at a final concentration of 35 μg/mL. Cells were grown in 7 × 2 L flasks with baffles containing 250 mL of deuterated medium (85% D), induced at OD_600_ of 0.6 using 1 mM of IPTG for 16 h at 37 °C. Cells were harvested by centrifugation and stored at −80 °C.

### 2.3. Protein Refolding and Purification

EDIII from *E. coli* was expressed in inclusion bodies as insoluble protein aggregates. The same refolding and purification protocol was used for both hydrogenated and deuterated proteins using hydrogenated buffers. Cells were thawed and lysed via the addition of 500 µL of 100 mg/mL lysozyme (62971-10G-F, Sigma-Aldrich, Poole, UK), 10 µL DNase I (EN0525, ThermoScientific 1 U/µL, Loughborough, UK), and protease inhibitors—10 mM of NaF, 1 mM of benzamidine hydrochloride, 1 mM of PMSF—and allowed to incubate at room temperature for two hours. Cells were then sonicated for 5 min (30 s on/off) on low setting (Biorupter™ UCD-200, New Jersey, NJ, USA) and the lysate were pelleted for one hour at 16,000× *g*. Inclusion bodies were washed three times with each buffer 1 (50 mM of NaH_2_PO_4_, 200 mM of NaCl, 5 mM of EDTA, 0.5 M of Urea, 1% Triton-X100, pH 6.0) and buffer 2 (50 mM of NaH_2_PO_4_, 1 mM of EDTA, 1 M of NaCl, pH 6.0) with intermittent sonication for 5 min and centrifugation at 16,000× *g* for 20 min. Purified inclusion bodies were resuspended in 10 mL of PBS, aliquoted to 1 mL, and frozen at −20 °C.

For protein refolding, inclusion bodies were thawed at room temperature and pelleted at 16,000× *g* for 20 min. EDIII was obtained via the solubilisation of inclusion bodies in solubilisation buffer (8 M of Urea in PBS) for four hours whilst shaking vigorously. Solubilised protein was diluted to 1mg/mL using solubilisation buffer and dialysed stepwise using 2 × 1L PBS. Refolded protein was centrifuged for 20 min at 5000× *g* to remove the precipitate. Protein was purified via immobilised metal ion affinity chromatography (IMAC) using Profinity resin (156-0123, Bio-Rad, Hemel Hempstead, UK) and eluted using an imidazole gradient consisting of a binding buffer (50 mM of NaH_2_PO_4_, 200 mM of NaCl, 10 mM of imidazole, pH 8) and elution buffer (50 mM of NaH_2_PO_4_, 200 mM of NaCl, 300 mM of imidazole, pH 7.75). Fractions were analysed by SDS-PAGE, and protein-containing fractions were pooled and concentrated using 3 kDa molecular weight cut-off concentrators (MAP003C36, Pall, Portsmouth, UK). Proteins were dialysed (D035, GeBAFlex, Generon, Slough, UK) into TEV cleavage buffer (50 mM of Tris, 0.5 mM of EDTA, 150 mM of NaCl, 1 mM of DTT, pH 8) and 10µl of TEV protease (T4455, Sigma-Aldrich, Poole, UK) per 1 mg of protein was added, followed by incubation overnight at 4 °C. His-tagged TEV was removed via IMAC using HisTrap™ (29-0510-21, Cytiva, Sheffield, UK), and fractions were analysed by SDS-PAGE. Fractions containing cleaved protein were concentrated and further purified via size exclusion chromatography (17-5174-01, Superdex™75 10/300 GL, Cytiva, Sheffield, UK; calibrant—#1511901, Bio-Rad, Hemel Hempstead, UK) using 50 mM of Tris, 150mM of NaCl, pH 7.5, as a running buffer. Protein-containing fractions were concentrated to up to 10 mg/mL.

### 2.4. Glycosaminoglycan’s (GAGs)

The glycosaminoglycans used in this study were: heparin calcium (H) (Lot–JC-11412@193, Celsus laboratories, Cincinnati, UK), chondroitin 6-Sulphate sodium salt from shark cartilage (CSCi) (27043, Biochemika, Buchs, Switzerland), chondroitin 6-sulfate sodium salt from porcine intestinal mucosa (CSCii) (Batch—YC314581701, Cas—12678-07-8, Carbosynth Limited, Compton, UK), dermatan sulphate (DS)—(lot—94604, Ronzoni Institute, Milan, Italy), Chondroitin Sulphate A (CSA) (YC15288, Carbosynth, Compton, UK), heparan sulphate (HS) (kind gift from M. Lima, Keele University, Keele, UK).

### 2.5. Circular Dichroism (CD)

Circular Dichroism (CD) was conducted on a Jasco J-1100 instrument available at the University of Liverpool. EDIII and GAGs were dialysed independently into three buffers (PBS at pH 5.5 and 7.5 and acetic acid buffer at pH 4.0) overnight at 4 °C. Protein and GAGs were combined in a 1:1 *w*/*w* ratio to give a final concentration of 3.1 mg/mL (pH 7.5) or 1.55 mg/mL (pH 5.5 and 4). The CD instrument was calibrated using camphorsulphonic acid prior to each experiment and buffer or buffer with GAG contributions subtracted for each sample. CD spectra were collected from 180 to 280 nm using a cuvette with a 0.2 mm pathlength (# 106-0.20-40, Hellma, Mühlheim, Germany). For wavelength spectra acquisition, fixed interval scanning was performed at every nm at 50 nm/min with 5 data acquisitions. CD data in the range of 180–250 nm were deconvoluted using BeStSel (Budapest, Hungary) [27]. Wavelength temperature interval data were collected between 25 and 80 °C in intervals of 1, 2, or 5 °C between wavelengths 180 and 250 nm every 0.1 nm. Then, 2D contour maps of the wavelength/temperature interval CD spectra were visualised using the OriginLab^®^ software (Northampton, MA, USA). For melting temperature determination, the wavelength/temperature interval data at 216 nm were fitted with a sigmoidal fit using SigmaPlot^®^ (San Jose, CA, USA). All the data were smoothed using the binomial method to five iterations, corrected for baseline deviations in the 250–260 nm region. The spectra from GAGs were deducted from EDIII with GAG spectra to visualise any changes to the EDIII wavelength spectra and for secondary structure deconvolution.

### 2.6. Heparin Affinity Column

An aliquot of EDIII at 100 µg/mL in PBS at pH 7.5 or 5.5 was loaded onto a 1 mL HiTrap heparin column (17-0407-01, Cytiva, Sheffield, UK) connected to a HPLC system (Cecil Instruments, Cambridge, UK) at a flow rate of 1 mL/min. The column was held under isocratic flow for 5 min to elute unbound material prior to the development of an increasing linear gradient of 0–2 M NaCl over 30 min. Elutants were detected by in-line monitoring at λ_abs_ = 280 nm and 214 nm.

### 2.7. Differential Scanning Fluorimetry (DSF)

EDIII was incubated with GAG samples in a 1:1 *w*/*w* ratio and dialysed into PBS at three pH values (7.5, 5.5, and 4.0) overnight. Experiments were carried out using the Protein Thermal Shift™ Dye Kit (Cat. 4461146, Applied Biosystems, Warrington, UK) following the manufacturer’s instructions and run on a StepOnePlus™ Real-Time PCR System (Applied Biosystems, Warrington, UK). Protein concentrations of either 2.5 or 5 µM were used.

### 2.8. Small Angle X-ray Scattering (SAXS)

SAXS was performed at the BM29 beamline (MX-1987, European Synchrotron Radiation Facility, Grenoble, France) and at the B21 beamline (SM-25074, Diamond Light Source Ltd., Didcot, UK). BM29 was equipped with a PILATUS 1 M detector (DECTRIS Ltd., Baden-Daettwil, Switzerland); the wavelength (λ) of the beam was 1 Å ranging from 0.004 to 0.493 Å^−1^. B21 was equipped with an Eiger 4 M detector (DECTRIS Ltd., Baden-Daettwil, Switzerland); the λ of the beam was also 1 Å, with q ranging from 0.003 to 0.44 Å^−1^. To account for concentration effects, the samples were subjected to a 1:2 dilution series immediately before bioSAXS analysis. Initial concentrations of between 2 and 8 mg/mL were used.

SAXS data were collected as 10 × 1 s exposure on 100 µL samples at 25 °C. A continuous flow cell capillary was used to reduce radiation damage. The data was averaged, the frames compared, and those displaying significant alterations discarded. SAXS profiles were subtracted by the corresponding blanks, and ScÅtter (Robert Rambo at Diamond Light Source, Didcot, UK, Bioisis.net), was used to estimate the radius of gyration (R_g_) in the Guinier Region where qR_g_ < 1.3. Plots were generated using Matplotlib [28]. The EDIII X-ray crystal structure (PDB 6FLB) was used to calculate the theoretical scattering using (PEPSI) modelling server [29]. Molecular weight was determined from *I* (0) using Guinier analysis.

### 2.9. Small Angle Neutron Scattering (SANS)

Small Angle Neutron Scattering (SANS) data were acquired on D11 at the Institut Laue-Langevin (ILL), Grenoble, France. A ^3^He MWPC (MultiWire proportional Counter) detector of 0.96 × 0.96 m^2^ was used, with 256 × 256 pixels. Samples were placed into quartz cells (110-QS, Hellma, Mühlheim, Germany) of 1 mm pathlength, with an illuminated cross-section of 7 × 10 mm^2^. A wavelength λ of 5.5 Å (relative fwhm 10%) was selected, and data were recorded at 3 configurations with sample-to-detector distances of 1.4, 5.5, and 8 m (collimation at 4, 5.5, and 8 m, respectively), covering a q-range of 0.009–0.42 Å^−1^, where q is the magnitude of the wave-vector (q = 4π/λ sin(θ/2), θ being the scattering angle). For selected samples, additional measurements were performed at the longest detector distance (39 m) to reach a lower q in order to confirm the intensity plateau obtained at higher q-values. Data were corrected for the relative pixel efficiencies, detector noise and dead-time, transmission, and incoming flux. The contribution of the empty cell was subtracted, and an absolute scale was obtained using the intensity of H2O (1 mm pathlength) as a secondary standard. The 2D data were finally azimuthally averaged. Data are available on demand (DOI:10.5291/ILL-DATA.8-03-946). The EDIII X-ray crystal structure 6FLB was used to calculate the theoretical scattering using (PEPSI) modelling server [29]. Molecular weight was determined from *I* (0) using Guinier analysis. Data were fitted using SASfit (Joachim Kohlbrecher, PSI, Villigen, Switzerland) [30].

### 2.10. Ab Initio Shape Determination

Pair distance distribution functions *P*(*r*) were calculated using ScÅtter IV [31]. All the ab initio shape modelling from the SANS data was conducted using DAMMIF [32], GASBOR [33], and DENSS [34]. The *P*(*r*) was calculated using ScÅtter IV [31] as an input for all algorithms. For DAMMIF, 20 iterations of the algorithm were performed with no symmetry or anisometry. For GASBOR, no symmetry or anisometry was inputted and 300 dummy resides were used.

## 3. Results and Discussion

### 3.1. Recombinant EDIII Is a Monomeric, Globular, and Well-Folded Protein

Previous studies have demonstrated that full-length Dengue envelope protein is a glycoprotein which can successfully be overexpressed and purified from Schneider 2 cells [35]. Its third domain, EDIII, is not thought to possess glycosylation sites, and can be expressed alone or as a correctly folded MBP fusion protein which displays antibody epitopes present on the virion [36,37,38,39]. Here, we successfully expressed and refolded EDIII as a recombinant, His_6_-tagged protein from *E. coli* to a high degree of purity (Appendix A). The preparation of inclusion bodies was followed by refolding via dialysis and subsequent nickel affinity chromatography. After the cleavage of the His_6_-tag using TEV, EDIII was purified via size exclusion chromatography. Expression yields of up to 162 mg of crude protein per litre of bacterial cell culture were achieved, which resulted in up to 29 mg of purified and cleaved EDIII. The monodispersity of the protein was confirmed by size exclusion chromatography with an apparent molecular weight of 11.4 kDa (Appendix A). Deuterated EDIII was purified in the same manner and yielded similarly pure protein with yields of up to 17.3 mg protein per litre bacterial flask cell cultures.

In order to further characterise the recombinant tag-free EDIII protein with regard to its proper folding and stability, we conducted experiments using CD (see Table 1) and DSF. Wavelength scans in CD were deconvoluted using BeStSel [27] and indicated that, at pH 7.5, the protein consists of 59.4% β-strand, 11.4% turn, and 39.2% unordered structural elements (Figure 1b). This is in agreement with the secondary structure observed in published structural data—e.g., PDB codes 6FLA, 6FLB, and 6FLC (Figure 1d). Slight discrepancies are expected due to PDB depositions representing EDIII bound to Fab or derived from full-length protein with X-ray crystallography or high-resolution cryo-EM data (e.g., PDB codes 3C5X, 1OAN, 1OK8, 4UIF, or 5A1Z), as opposed to the unbound, in situ-derived data of the EDIII domain only, which we present here. Very little change in secondary structure was observed when EDIII was measured at pH 7.5, 5.5, and 4 at room temperature, suggesting that the protein is stable over a range of biologically relevant pH and does not alter its conformation greatly (see Figure 1a,b). Upon the deconvolution of the CD wavelength spectra from pH 5.5, a small contribution from α-helix (6%) was calculated, however we believe this to be due to BeStSel not being able to fit the CD spectra with precision due to the large maxima at 225nm. The analysis of several different published structures (e.g., 3C5X, 1TG8, 1OK8, and 1OAN) does not show any α-helical content within the EDIII domain.

Notably, the CD wavelength spectrum shows a marked maximum between 221 and 241 nm which can be attributed to the cysteine bridge formed between Cys302 and Cys333 in EDIII. The maximum at ~230 nm made deconvolution using conventional programs difficult, resulting in some fits presenting higher normalised root-mean-square deviation (NRMSD) values and leading to a small degree of uncertainty with regard to the secondary structure prediction using BeStSel [27] (ranging from 0.0517 to 0.13067—see Appendix A). This is also probably why there is a small component from the α-helices at pH 5.5.

To further assess the stability and fold of the protein, we conducted thermal denaturation experiments in CD (Figure 1c and Table 1) and using DSF. Both experiments resulted in two-state transition curves with a melting temperature of 52.4 °C in CD at pH 7.5 (Table 1 and Appendix A), which was corroborated with DSF at 52.3 °C (Appendix A). These results indicate a stable, folded recombinant protein. Thermal denaturation experiments conducted using CD showed the greatest change in CD signal between 205 and 220 nm, indicative of the unfolding of the protein (Figure 1c).

Another change was observed for the maximum at ~230 nm, which disappeared at around 45 °C, which is a slightly lower temperature than the complete unfolding of the protein. The reduction of the disulfide bridge in EDIII led to a decrease in the temperature at which this maximum disappeared with increasing DTT concentrations (Appendix A), suggesting that the maximum at ~230 nm is attributable, at least in part, to the disulphide bridge present in EDIII. Typically, disulphide bridges show a maxima in CD around 250–270 nm [40], however our samples may also experience significant contributions of aromatic residues distorting the CD spectrum within the far-UV range. Tryptophan, in particular, has been shown to influence the far-UV CD spectra of proteins with minimal α-helical content [41]. At a size of 11 kDa, our protein contains one tryptophan, four phenylalanines, and three tyrosines, as well as displaying minimal α-helical components. Future mutational studies will confirm the contributions of individual residues to the CD spectrum.

Both the SANS and SAXS spectra of EDIII are consistent with those predicted from the polynomial expansions of protein structures and interactions (PEPSI) modelling server [29] for a well folded, monomeric, and globular protein (Figure 2a,b). The radius of gyration (R_g_) is consistent for experimentally derived data from both SAXS and SANS at 14.4 and 15.5 Å, respectively. These values are in the same range as those obtained from simulated data (13.2 and 14.0 Å for SAXS and SANS, respectively) using the PEPSI server, corroborating our previous observations of a globular shape for EDIII. This is further supported by Kratky analysis, which shows a typical profile for a well-folded globular protein (Figure 2a inset). Approximations of molecular weights from the SAXS and SANS data were in good agreement with the values obtained from size exclusion chromatography (11.4 kDa) and with the theoretical value of 11.18 kDa. Additionally, there is a discrepancy between high-q SANS and the simulated data in the form of a “bump” (q = 2–4 nm^−1^), indicating a structural difference in the sub-nanometre scale. The low-resolution ab initio shape modelling of EDIII at pH 7.5 and 5.5 using DAMMIF [32] showed subtle differences in the tertiary structure of the protein, changing from a “dumbbell-like” shape at pH 7.5 to a “croissant-like” shape at pH 5.5 (Figure 2c,d). Additionally, EDIII at pH 7.5 is represented with slightly larger dimensions than EDIII at pH 5.5, which is also reflected in the differing R_g_ values and suggests a slightly “swollen” protein molecule at pH 7.5. The placing of the X-ray structure of EDIII (PDB code 6FLA) into the shape envelope indicates that these pH-dependent subtle changes are most likely due to dynamic changes in the loop regions connecting the ß-stranded core structure. Previous studies support this notion that the viral envelope protein is sensitive to pH changes and in fact exploits this effect in order to adopt the specific conformations required at different stages of endocytosis [42,43].

### 3.2. Interactions with Heparin and Chondroitin Sulphate C Are pH-Dependent and Cause Changes at a Molecular Level

Different flaviviruses have been shown to preferentially bind to certain GAGs. Zika envelope protein prefers longer heparin oligosaccharides [6] whereas Japanese Encephalitis Virus (JEV) requires highly sulphated GAGs, heparin and dextran sulphate to successfully attach to and infect BHK-21 cells [44]. Here, we conducted a systematic study of the binding of different GAGs to 6xHis-tagged EDIII using biophysical techniques in order to understand the interaction at a molecular level at three Ph–pH 4 (in acetate buffer), and pH 5.5 and 7.5 (in PBS).

At both pH 7.5 and 5.5, 6xHis-tagged EDIII differed from cleaved EDIII in its α-helix and anti-parallel β-sheet components, with 6xHis-tagged EDIII having approximately 12–14% more α-Helix, and ~14% less anti-parallel β-sheet content (compare Figure 1b and Figure 3c,d). This shift in proportion of secondary structure is probably caused by the additional N-terminal 6x-His-Tag and linker region in our construct, which comprises 45 additional amino acids as compared to the cleaved EDIII construct.

Recombinantly generated 6xHis-tagged EDIII protein was incubated with equal amounts of different GAGs (*w*/*w*)—namely heparin (H), dermatan sulphate (DS), chondroitin sulphate A (CSA), chondroitin sulphate C from shark cartilage (CSCi), and chondroitin sulphate C from porcine intestinal mucosa (CSCii), dialysed into the appropriate buffer, and changes in the secondary structure were analysed using CD. Samples incubated in PBS at pH 5.5 and 7.5 gave good wavelength spectra; however, EDIII incubated with GAGs at pH 4.0 in acetate buffer precipitated out of solution. Neither EDIII nor the GAGs precipitated out of solution when exposed to this buffer independently.

The wavelength spectra of EDIII with GAGs at pH 7.5 and 5.5 (Figure 3a,b) showed minor differences with regards to their appearance and location of minima and maxima. All the wavelength spectra showed a maximum at 225 nm and minima at 210 and 195 nm to varying degrees. Wavelength spectra were deconvoluted using BeStSel [27] (Figure 3c,d), and slight differences in secondary structure contents were observed in addition to the increase in α-helical content in the his-tagged EDIII, as discussed earlier.

A comparison of the deconvolution of spectra obtained at pH 7.5 and 5.5 with different GAGs (Figure 3c,d) shows a subtle increase in the anti-parallel β-sheet content at pH 7.5 with CSCi and H. This may suggest that CSCi and H interact with EDIII to induce dynamic changes in its secondary structure in a pH-dependent manner. However, when attempts were made to further define the interaction between EDIII and H using a heparin affinity column, EDIII was eluted isocratically prior to the commencement of a linear gradient of NaCl, indicating that any changes heparin may induce in EDIII are subtle (Appendix A). This effect was greater at pH 5.5 compared to pH 7.5.

Furthermore, we examined changes occurring at the molecular level upon GAG binding using SANS and SAXS. SANS data were obtained in matchout conditions for GAGs at concentrations of 2mg/mL, therefore only the protein contributed to scattering [45], whereas at 30 mg/mL of scattering was observed for CSCi and H in pure D_2_O and is reported in the Appendix A. This is in contrast to SAXS, where contributions to the scattering curves were equivalent for GAGs and EDIII.

The SANS and SAXS spectra show changes at the molecular level when EDIII was incubated with H and CSCi at pH 7.5, as evidenced by a slower intensity decay (lower Porod exponent) in the high-q regions observed in SANS for EDIII with H, and a “bump” in both SAXS and SANS for CSCi at pH 7.5 (Figure 4a,b, Appendix A). This change was not observed at pH 5.5 (Figure 4c,d). Interestingly, the interaction of EDIII with H and CSCi resulted in distinctly different appearances of the two curves at high-q, indicating different changes at the molecular level at pH 7.5, and may be representative of different binding modes and/or interaction interfaces with EDIII.

The fitting of the SANS data was performed with a model of generalised linear polymer coils [46] using SASfit [30]. This model, developed for homopolymers, allows to reproduce the smooth decay of intensity observed at high q, obtaining a so-called “Flory” parameter – later denoted υ (Nu)- that can be interpreted as a degree of compactness of the protein. The fit also provides a gyration radius R_g_ and a forward scattering that is proportional to the molecular weight and concentration of protein, or protein aggregate. For most samples, the values are in the range expected for a dense or compact structure (see Appendix A). However, EDIII with H indicates a complete loss of compactness, while Guinier analysis revealed an unchanged R_g_, indicating that H induces a change in the conformation of EDIII from a compact, globular molecule to a slightly extended conformation with a more flexible chain. This corroborates our earlier findings from CD that heparin has a destabilising effect on the secondary structural elements of EDIII at pH 7.5 but not pH 5.5 (Figure 3), thus preventing EDIII from forming a compact globular molecule (Table 1, Appendix A). However, no changes to the calculated molecular weight were observed with SAXS, indicating that whilst heparin is capable of inducing some secondary changes to EDIII, it seems to lack the ability to strongly bind EDIII. This may indicate that the efficient binding of heparin to EDIII also requires the presence of additional motifs in EDI and EDII, which are not present within our construct [6]. This is in agreement with the findings from the heparin affinity assay, which showed that EDIII eluted isocratically prior to the NaCl gradient.

Nu and Guinier analysis of SANS data for EDIII with CSCi indicate negligible changes to the compactness of EDIII but with a significant increase in R_g_, comparable to that of multimer formation, indicating that whilst EDIII remains relatively unchanged at the molecular level (as corroborated by the findings from CD), CSCi induces higher order molecular changes, which again are pH-dependent. The pH dependency of the binding of CSCi to EDIII raises the intriguing possibility that CSCi may bind EDIII and thus be involved in endocytosis at the pre-fusion stage at pH 7, and that upon the induction of a localised drop in pH, the viral protein is released from the endosome to facilitate viral absorption [8,47].

HS, DS, CSA, and CSCii did not cause differences in the SANS or SAXS spectra of EDIII at both pH 7.5 and 5.5 and were therefore not considered as strong binders for EDIII. Furthermore, no aggregation was observed in any of the samples, as evidenced by the formation of a plateau at the low-q range. There were no visible effects of pH in those samples, as we noted: there is excellent agreement between the SAXS and SANS curves at different pHs (Figure 3b,d and Figure 4a,c). We observe a small difference in the CSCii sample at a high-q between the two pH, however we believe this to be an artefact of the data collection.

We further note that when comparing the samples of EDIII with CSCi and CSA or EDIII with H and HS, respectively, the spectra for SAXS and SANS are different, which might indicate that any interaction is likely dependent on sulphation state of the GAG. Different sulphation states and carboxylic acid positioning between GAGs would result in different charge maps for the molecules, whereas changes in the linker position would result in different overall shapes of GAG molecules. This would explain why some GAGs are able to interact with EDIII whilst others of a similar primary composition cannot—i.e., CSA vs. CSCi and, to some extent, H vs. HS.

### 3.3. Interaction with Chondroitin Sulphate C Indicates Higher-Order Molecular Organisation

SANS data collected for EDIII with CSCi showed differences at high-q (q = 1.5 to 4 nm^−1^) compared to EDIII only, which indicates changes in EDIII at the molecular level. In addition, the SANS data also showed a significant change in curvature at mid- and low-Q ranges (Figure 4b), indicating an increase in particle size. The determination of R_g_ reveals an approximate diameter of 120–140 Å, and the analysis of forward scattering indicated a molecular weight approximately three times higher than monomeric EDIII, approximating 35 kDa. Subsequent ab initio shape determination using GASBOR [33], DAMMIF [32], and DENSS [34] all returned an elongated structure with an irregular mass distribution with a larger, denser mass on one side and a somewhat smaller mass on the other. Although the mass density seems to be linked in the models, we interpret the modelling results as three to five separate EDIII monomers being brought into special proximity by a CSCi molecule, resembling “beads on a string” (Figure 4e, Appendix A). The connection of the separate EDIII masses in the models is probably a limitation of the modelling software treating the EDIII:CSCi complex as one continuous shape. Although our results provide strong evidence for a CSCi-mediated complex formation of multiple EDIII monomers, our data do not reveal directly how CSCi coordinates multiple molecules of EDIII. Whether CSCi linearly binds the same interface of each EDIII molecule or whether CSCi wraps around the protein molecules remains to be determined.

Commercially supplied GAGs (usually purified from natural sources) are polydisperse mixtures of varying chain length with a repeating motif throughout the chain. Typically, the molecular weights of CS samples display considerable variation and may range from lower weight species of around 10 kDa to higher molecular weight species of up to 100 kDa [48]. The distribution of molecular weights is dependent upon various factors, including species of origin, purification methodology, and type of CS. The molecular weight of CSC purified from shark cartilage, as similar to the samples used in our study, ranges from 50 to 70 kDa [49]. One CSC disaccharide subunit has a molecular weight of around 500 Da, which would give a chain length of approximately 120–140 disaccharides. Therefore, we suggest that CSCi is likely to be capable of binding two to three molecules of recombinant EDIII as represented by the denser component of the model. Three to five EDIII molecules may bind a single polysaccharide chain at various interaction points along the chain, with the number of EDIII molecules bound dependent on the length of the chain. The number of monomers bound to the chain would be dependent upon two factors—the number of recognition points along the polysaccharide chain, and steric constraints which arise from the size of the monomer. Considering our ab initio models, longer chains would be able to bind more EDIII molecules, as they are present in a comparatively lower amount and would therefore contribute less scattering to the SANS data. This is likely represented by the less dense components of the models (Figure 4e, Appendix A). As a result, our ab initio models represent with a larger, denser component on one side and a less component on the other extending away from the dense component. Given the steric constraints and physical spacing between EDIII molecules on the capsid envelope, it is possible that a single CSCi molecule does not bind multiple EDIII molecules during viral endocytosis as we observed. CSCi binding multiple EDIII molecules may be a result of our protein’s much smaller size in comparison to the whole Dengue virion and the repeating disaccharide motif of CSCi. Nevertheless, our findings show that CSCi is capable of binding EDIII and implicates it in the viral endocytosis process, potentially as a co-receptor preceding endocytosis.

## 4. Conclusions

There is a growing body of evidence of the role of GAGs as cofactors for several viruses such as Zika [6], Dengue, Chikungunya [50], and Japanese encephalitis virus [51]; however, the extent of their involvement in viral endocytosis is yet to be fully understood.

Our systematic study of the interactions between EDIII and a range of biological relevant GAGs revealed that both CSCi and H bind to recombinantly generated EDIII in a pH-dependent manner while causing subtle changes to its secondary structure. However, EDIII remains an overall globular protein. Interestingly, heparan sulphate, a lesser sulphated derivative of heparin, was unable to induce the secondary structure changes observed with heparin, demonstrating the importance of sulphation state and sulphate group positioning as a prerequisite for ensuring specific charge density and distribution on the polysaccharide, which in turn mediate specific interactions between GAGs and viral envelope proteins. Our findings contradict previously published data in which heparin has been shown to bind with a strong affinity to the full-length DENV envelope protein [6,13]. However, our study was conducted using a truncated EDIII construct, and it is therefore possible that other components of the viral envelope protein are required to mediate this binding which are apparently lacking in our EDIII construct.

Experiments conducted by SAXS and SANS indicated the formation of a larger EDIII oligomer in the presence of CSCi at pH 7.5. The ab initio shape determination of SANS data revealed a linear arrangement of three to five EDIII molecules with varying density, whose exact oligomeric structure is likely dependent on the individual chain length of CSCi and resulting spatial constraints. GAGs involved in initial receptor identification may be capable of binding multiple copies of the viral E-protein prior to endocytosis, with the steric bulk of the virion and spatial distancing of the envelope proteins on the virus capsid [8] acting as limiting factors for how many binding events may occur for a given GAG molecule. However, our model lacks information about the interaction of EDIII with CSCi at the molecular level, which could reveal why EDIII shows a greater affinity for CSCi than other GAGs tested in our study. Interestingly, different strains of DENV have previously been found to differ in their affinity for various GAGs through the selective mutation of their envelope protein and glycosylation at specific residues, resulting in alterations in its charge density distribution [6]. This resulted in an altered receptor and co-receptor preference, thus accounting for its broad viral tropism.

Human CSC is located in various biological tissues including the skin, bone, cartilage, nerve tissues, and notably blood vessels [52]. Furthermore, CSC is present in the extracellular matrix of host organ stroma, which mostly consists of fibroblasts and pericytes. Pericytes, in particular, coat the endothelial cells lining the capillaries in humans [53]. The biological significance of the interaction of EDIII with CSCi raises the intriguing prospect of a role for CSC in the endocytosis of EDIII into vascular tissues. Similar interactions between chondroitin sulphate moieties and viruses have previously been observed [54,55].

Taken together, our observations reveal pH-dependent dynamic changes to the conformation of EDIII induced by both CSCi, and H, thus positioning GAGs as attractive therapeutic targets for the development of transmission blocking vaccines.

## Figures and Tables

**Figure 1 pathogens-09-00935-f001:**
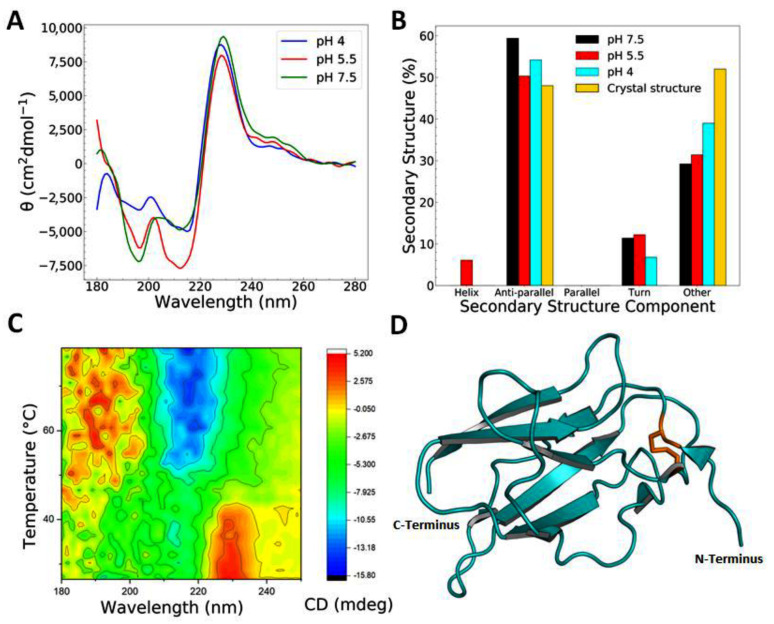
EDIII is well folded over a range of pHs. (**A**) Circular dichroism (CD) spectra of the third domain of the Dengue viral envelope protein (EDIII) at different pHs showing a characteristic maximum at around 230 nm. CD spectrum in reducing conditions lacks this peak. (**B**) Secondary structure determination from single CD spectrum analysis of tag-free EDIII at pH 4.0, 5.5, and 7.5 using BeStSel [27] and comparison to secondary structure components found in the crystal structure (Protein databank (PDB) codes 6FLA, 6FLB, and 6FLC, averaged). (**C**) A 3D plot of changes in the CD signal during the thermal denaturation of EDIII at pH 7.5. (**D**) Structural representation of EDIII (PDB code 6FLC) showing the all-beta fold of the protein. The disulphide bridge between cysteine residues is indicated as orange sticks.

**Figure 2 pathogens-09-00935-f002:**
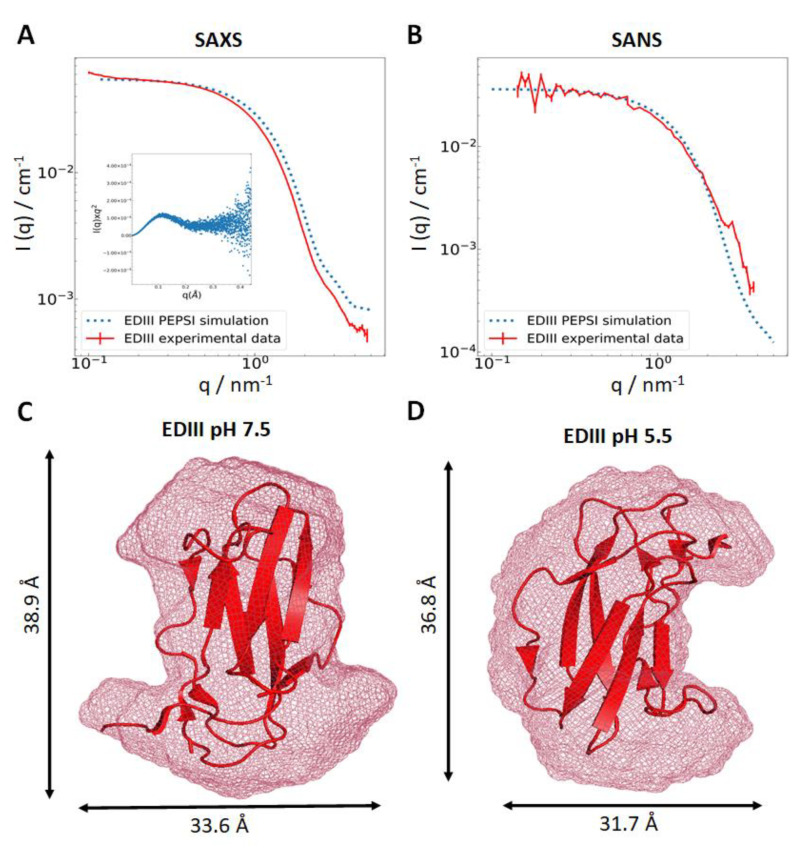
Scattering data and ab initio modelling using DAMMIF. Experimentally derived small angle x-ray scattering (SAXS) (**A**) and small angle neutron scattering (SANS) (**B**) data for EDIII plotted (in red) against simulated data from the polynomial expansions of protein structures and interactions (PEPSI) server (blue dots) [29], normalised Kratky analysis displayed as inset. PEPSI modelling was carried out using PDB code 6FLB (with Fab removed). Ab initio shape models of EDIII at pH 7.5 ((**C**), χ^2^ = 2.46) and 5.5 ((**D**), χ^2^ = 1.995) from the SANS data are represented as a light red mesh (modelled with DAMMIF [32]) with a high-resolution X-ray diffraction structure (PDB code 6FLA) manually placed within.

**Figure 3 pathogens-09-00935-f003:**
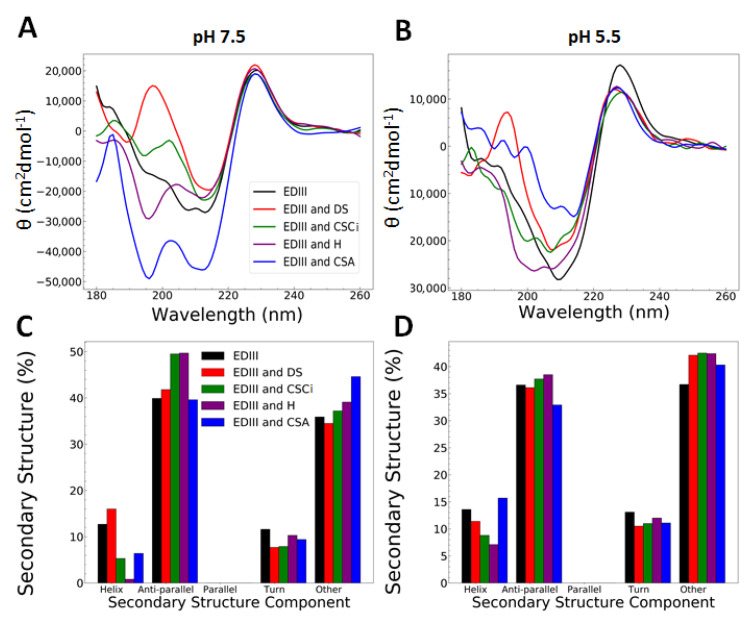
CD analysis of 6xHis-tagged EDIII post incubation with glycosaminoglycans (GAGs). (**A**,**B**) Wavelength spectral analysis of EDIII with dermatan sulphate (DS), Chondroitin sulphate C from shark cartilage (CSCi), heparin (H), and chondroitin sulphate A (CSA) pH 7.5 (**A**) and pH 5.5 (**B**). (**C**,**D**) Deconvolution of EDIII wavelength spectra with GAGs at pH 7.5 (**C**) and pH 5.5 (**D**) using beta structure selection server (BeStSel) [27].

**Figure 4 pathogens-09-00935-f004:**
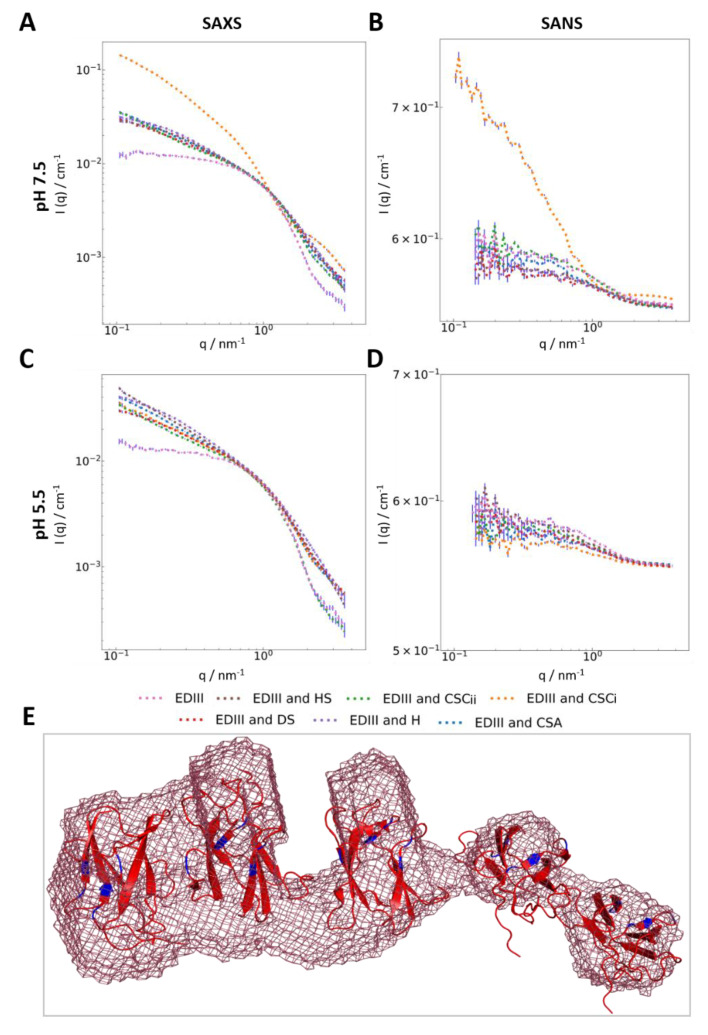
Scattering data for EDIII with GAGs at pH 7.5 and 5.5, and modelling of EDIII with CSCi. (**A**) SAXS data of EDIII with GAGs at pH 7.5, (**B**) SANS data of EDIII with GAGs at pH 7.5, (**C**) SAXS data of EDIII with GAGs at pH 5.5, (**D**) SANS data of EDIII with GAGs at pH 5.5, after subtraction of background, empty cell, and incoherent scattering. Error bars shown in blue. (**E**) Ab initio modelling of low-resolution SANS data of EDIII with CSCi at pH 7.5 using DAMMIF, χ^2^ = 0.9527 [32] represented as a light red mesh with X-ray structures of EDIII monomers (PDB code 6FLA) placed in areas of larger electron density. EDIII residues previously identified as potential GAG binding sites are shown in blue [6].

**Table 1 pathogens-09-00935-t001:** Comparison of the melting temperatures Tm (°C) derived from the sigmoidal fitting of CD data.

Sample	pH 7.5	pH 5.5
EDIII	52.38	50.9
EDIII & H	n/a	50.9
EDIII & CSCi	53.12	49.5
EDIII & CSCii	55	59.8

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
