# Peer review of "Molecular Changes in Dengue Envelope Protein Domain III upon Interaction with Glycosaminoglycans"

_pathogens, 2020, doi:10.3390/pathogens9110935_

Round 1
Reviewer 1 Report
Hyatt et al described the changes in envelope protein domain III when interacts with GAG using several biophysical assays including CD and SAXS showing the molecular changes to envelope protein.
Overall the manuscript is very well written and described. I have only minor comments as following
- Could author provide more suggestion as to why additional helix is observed in figure 1B from EDIII but not from crystal structure?
- Does the author have data for EDIII-CSCi interaction at low pH? As this can provide more insight information regarding the endocytosis.
Specific common
Line69 : please describe ‘kl’
Author Response
The authors thank the reviewer for reviewing our work and for their helpful suggestions. We have addressed these as follows:
- Could author provide more suggestion as to why additional helix is observed in figure 1B from EDIII but not from crystal structure?
Response - we propose the observed α-helical content of 6% (around 5-6 amino acids) at pH 5.5 is a result of BeStSel struggling to deconvolute the spectra with prevision owing to the maximum at 225nm. This has been noted in text. Other secondary structure deconvolution software’s such as dichro-web were unable to deconvolute the spectra due to this maximum.
- Does the author have data for EDIII-CSCi interaction at low pH? As this can provide more insight information regarding the endocytosis.
Response - We tried to conduct analysis at pH 4 also, however when the protein and GAG were combined in acetic acid buffer at pH 4, they immediately precipitated out of solution, making it impossible to conduct analysis. We have spoken to colleagues who have observed a similar effect when combining proteins and GAGs in acetic acid buffer. The reasons for this are currently unknown, however it is something that we are keen to research further. We discarded the samples and continued with the experiments for pH 5.5 and 7.5, however we plan to conduct further analysis at lower pH’s utilising alternative buffering systems.
Specific common
Line69 : please describe ‘kl’
Response - ‘kl’ was referring to the loop region between β-sheets ‘k’ and ‘l’ but has now been defined explicitly in the text.
We hope that these changes and explanations find your approval.
With kind regards,
Anja Winter
Reviewer 2 Report
The manuscript from Hyatt et al. report a systematic study investigating the effect of a range of biologically relevant glycosaminoglycans (GAGs) on the structure and oligomeric state of recombinantly generated the third domain of the viral envelope protein (EDIII) of dengue virus (DENV). Their systematic study of the interactions between EDIII and a range of biological relevant GAGs revealed that both CSCi and H bind to recombinantly generated EDIII in a pH-dependent manner while causing subtle changes to its secondary structure. This is a fairly professional manuscript. The authors explained in the manuscript the reasons and feasibility of using only recombinant EDIII fragments for experiments (also refer to References 34-38). These instructions and references can dispel readers' doubts about whether the use of prokaryotes to produce recombinant proteins and the use of short peptides can exhibit the true conformation in this study. The experiment performed well, the conclusions were correct, and the author also provided a lot of supplementary data.
My question is: in Supplementary Figure 3, it can not be seen that the molecular weight of TEV-cleaved EDIII is 11.4 kDa.
Author Response
The authors thank the reviewer for reviewing our work and their very valid point made. We would like to submit the following response to their query:
Gel filtration spectra has been substituted for a sample spectrum in which the column and running buffers were assessed with four SEC protein markers to calculate apparent molecular weight and had accompanying SDS-PAGE gel analysis of fractions. Previous spectrum has no corresponding protein marks and had no accompanying SDS-PAGE gel analysis of fractions.
We hope that this finds the reviewer's approval.
With kind regards,
Anja Winter
Reviewer 3 Report
Glycoamyloglycans (GAGs) have been shown to play a role in the initial viral attachment through interaction with domain III of the envelope protein (EDIII). In this paper, Hyatt et al. have done a systematic study of the interaction between EDIII and several different biologically relevant GAGs to study the dynamic changes that occur upon binding by using circular dichroism and small-angle scattering (SAXS and SANS). The authors reported the binding of GAGs to EDIII induced distinct dynamic changes at a molecular level. Heparin induced changes to the secondary structure of EDIII, whereas chondroitin sulfate C induced changes not only at the molecular level but also at the super-molecular level. The pH and sulfation state of GAGs also affect the interaction with EDIII. The authors also showed CSCi has the highest affinity among the GAGs tested, and one single polysaccharide chain of this GAG has the ability to bind to three to five EDIII molecules.
Minor comments:
Line 232: The Supplementary Figure 3B needs molecular weight markers to show the molecular weight of the EDIII.
Line 240: There are several high resolution cryoEM structures of dengue serotype 2 and also crystal structures of the envelope protein without Fab (unbound state).
Line 248: abbreviation NRMSD has not been described anywhere in the manuscript.
Line 263: In supplementary figure 4 (figure legend), it is the data of EDIII with heparin at pH 7.5, not 5.5, that could not be fit.
Line 292: The q values here (in 1/A) have different units than those in figure 2A-B (in 1/nm). It would be easier to read if they have the same unit. This is also for figure 4.
Line 299: There are several crystal structures of envelope protein and EDIII of flaviviruses that were solved at different pH. You could superimpose those structures to show the flexibility/dynamic of the loop regions.
Line 315: what is the reason for using 6xHis-tagged EDIII, not the untagged one, in this experiment?
Line 338: Heparin has been shown to have a helical structure (PDB ID: 1HPN). Since the 6xHis-tagged EDIII was incubated with equal amounts of GAGs, will the secondary structure of the GAGs (if there is, such as what heparin showed) affect the CD spectra of EDIII-GAG complexes?
Line 342: “…indicating that any changes heparin may induce in EDIII are subtle…”. EDIII was eluted isocratically before a linear gradient of NaCl because it requires other parts of E protein for a strong binding, just like what mentioned in line 377.
Line 343: Supplementary figure 7 shows the heparin chromatography runs at pH 4.5 and 5.5, not pH 5.5 and 7.5.
Line 358: “….regions observed in SANS for EDIII with H”, it should be SAXS, not SANS.
Author Response
The authors thank this reviewer for their very thorough reading of our manuscript and their very helpful suggestions. The manuscript was much improved as a result. We have addressed the reviewer's concerns as follows:
Line 232: The Supplementary Figure 3B needs molecular weight markers to show the molecular weight of the EDIII.
Response - Gel filtration spectra has been substituted for a sample spectrum in which the column and running buffers were assessed with four SEC protein markers to calculate apparent molecular weight and had accompanying SDS-PAGE gel analysis of fractions. Previous spectrum has no corresponding protein marks and had no accompanying SDS-PAGE gel analysis of fractions.
Line 240: There are several high resolution cryoEM structures of dengue serotype 2 and also crystal structures of the envelope protein without Fab (unbound state).
Response - Has been addressed in text.
Line 248: abbreviation NRMSD has not been described anywhere in the manuscript.
Response - NRMSD has been defined in text.
Line 263: In supplementary figure 4 (figure legend), it is the data of EDIII with heparin at pH 7.5, not 5.5, that could not be fit.
Response - This has been corrected.
Line 292: The q values here (in 1/A) have different units than those in figure 2A-B (in 1/nm). It would be easier to read if they have the same unit. This is also for figure 4.
Response - Angstrom has been changed to nm.
Line 299: There are several crystal structures of envelope protein and EDIII of flaviviruses that were solved at different pH. You could superimpose those structures to show the flexibility/dynamic of the loop regions.
Response - It is a good suggestion, but we chose to redirect readers to articles that can demonstrate this phenomenon and have investigated it in greater detail than we have in our publication. We have included more citations to this effect in line 299.
Line 315: what is the reason for using 6xHis-tagged EDIII, not the untagged one, in this experiment?
Response - Due to technical challenges at the time these experiments were conducted, his-tagged EDIII was initially used. We were limited with the amount of cleaved sample, so we decided to group experiments according to which samples needed to be compared with each other. We used the uncleaved EDIII for the GAGs CD experiments, and the cleaved EDIII for the pH CD experiments. We discovered that the his-tag only became a problem in SAXS and SANS data because they were visible in their spectra so decided to utilise cleaved sample for those experiments. We are aware that there is a slight increase in α-helical content when comparing cleaved and uncleaved EDIII which is why we decided to only compare samples with each other where the same EDIII has been used (cleaved or non-cleaved).
Line 338: Heparin has been shown to have a helical structure (PDB ID: 1HPN). Since the 6xHis-tagged EDIII was incubated with equal amounts of GAGs, will the secondary structure of the GAGs (if there is, such as what heparin showed) affect the CD spectra of EDIII-GAG complexes?
Response - We ran equivalent amounts of GAGs in the same buffer in CD and subtracted the spectra from the GAG and EDIII spectra, so what was left would be EDIII and any induced changes from the presence of the GAGS, and we compared this to EDIII alone to see if there was a difference. We ensured that only either the buffer was subtracted for the EDIII samples alone, or the buffer with GAG contributions was subtracted from the EDIII with GAG samples so that the buffer was not subtract twice. Mark Skidmore said that the concept of GAGs having secondary structure (in solution) has been refuted in the field some 10-15 years ago and that GAGs don't usually have signal in CD unless they're above 10mg/ml. We are aware that there is at least one X-ray crystal structure available showing a helical-like conformation of a heparin oligosaccharide, but this is most likely due to induced conformation upon binding to the protein it is co-crystallised with.
Line 342: “…indicating that any changes heparin may induce in EDIII are subtle…”. EDIII was eluted isocratically before a linear gradient of NaCl because it requires other parts of E protein for a strong binding, just like what mentioned in line 377.
Response - Comments in line 377 have been amended to reflect this and link back to the heparin column.
Line 343: Supplementary figure 7 shows the heparin chromatography runs at pH 4.5 and 5.5, not pH 5.5 and 7.5.
Response - The Heparin columns were run at pH 5.5 and 7.5. We have conferred with our co-authors and they have confirmed this. Labelling has been amended.
Line 358: “….regions observed in SANS for EDIII with H”, it should be SAXS, not SANS.
Response - We have double checked our data, and confirmed a very small bump with EDIII and heparin in SANS but not SAXS. To make this point very clear to readers of the journal, we decided to include an additional figure in supplementary material to show the data in more detail.
We hope that we have addressed the reviewer's concerns sufficiently.
With kind regards,
Anja Winter